# L-Lysine from *Bacillus subtilis* M320 Induces Salicylic-Acid–Dependent Systemic Resistance and Controls Cucumber Powdery Mildew

**DOI:** 10.3390/ijms26146882

**Published:** 2025-07-17

**Authors:** Ja-Yoon Kim, Dae-Cheol Choi, Bong-Sik Yun, Hee-Wan Kang

**Affiliations:** 1Division of Horticultural Biotechnology, School of Biotechnology, Hankyong National University, Anseong 17579, Republic of Korea; xhfhcl@naver.com; 2Division of Biotechnology and Advanced Institute of Enviroment and Bioscience, College of Enviromental and Bioresource Sciences, Jeonbuk National University, Iksan 54596, Republic of Korea; ell612@naver.com (D.-C.C.); bsyun@jbnu.ac.kr (B.-S.Y.); 3Institute of Genetic Engineering, Hankyong National University, Anseong 17579, Republic of Korea

**Keywords:** *Bacillus subtilis* M320, L-lysine, cucumber, powdery mildew, systemic acquired resistance, salicylic acid

## Abstract

Powdery mildew caused by *Sphaerotheca fusca* poses a significant threat to cucumber (*Cucumis sativus* L.) production worldwide, underscoring the need for sustainable disease management strategies. This study investigates the potential of L-lysine, abundantly produced by *Bacillus subtilis* M 320 (BSM320), to prime systemic acquired resistance (SAR) pathways in cucumber plants. Liquid chromatography–mass spectrometry analysis identified L-lysine as the primary bioactive metabolite in the BSM320 culture filtrate. Foliar application of purified L-lysine significantly reduced powdery mildew symptoms, lowering disease severity by up to 92% at concentrations ≥ 2500 mg/L. However, in vitro spore germination assays indicated that L-lysine did not exhibit direct antifungal activity, indicating that its protective effect is likely mediated through the activation of plant immune responses. Quantitative reverse transcription PCR revealed marked upregulation of key defense-related genes encoding pathogenesis-related proteins 1 and 3, lipoxygenase 1 and 23, WRKY transcription factor 20, and L-type lectin receptor kinase 6.1 within 24 h of treatment. Concurrently, salicylic acid (SA) levels increased threefold in lysine-treated plants, confirming the induction of an SA-dependent SAR pathway. These findings highlight L-lysine as a sustainable, residue-free priming agent capable of enhancing broad-spectrum plant immunity, offering a promising approach for amino acid-based crop protection.

## 1. Introduction

Powdery mildew, caused by the fungal pathogens *Podosphaera xanthii* and *Sphaerotheca fusca*, are among the most destructive diseases affecting cucumber cultivation worldwide [1]. These pathogens produce abundant spores that disperse rapidly under favorable environmental conditions, leading to substantial yield losses and deterioration in fruit quality. Although chemical fungicides are commonly used to manage powdery mildew outbreaks, their frequent application has raised significant concerns regarding pesticide residues, environmental contamination, and the emergence of fungicide-resistant pathogen populations [2,3]. Consequently, there is a growing need to develop sustainable, environmentally friendly strategies to effectively manage this disease.

Biological control using antagonistic microorganisms, particularly *Bacillus* spp., has emerged as a promising alternative. *Bacillus subtilis* strains are known for their ability to produce diverse antimicrobial metabolites, including cyclic peptides (iturins), cyclic lipodepsipeptides (surfactins and fengycins), and various other physiologically active compounds [4,5,6]. These metabolites can directly inhibit pathogens or indirectly enhance plant health. Furthermore, recent studies suggest that certain *Bacillus* strains can also stimulate plant immune responses by inducing systemic acquired resistance (SAR) or induced systemic resistance (ISR), thereby providing durable protection against a broad spectrum of pathogens [5,7].

Beyond their well-established roles as building blocks of proteins, amino acids have emerged as critical regulators of plant immunity. Recent studies have demonstrated that amino acids actively participate in a wide range of physiological and biochemical processes, including hormonal homeostasis, carbon/nitrogen (C/N) balance, and the biosynthesis of secondary metabolites crucial for plant defense responses [8]. Notably, pathogen infection often leads to substantial increases in amino acid concentrations within plant tissues, suggesting their involvement in modulating immune signaling pathways. Among these, lysine-derived pipecolic acid has been identified as a key signaling molecule that amplifies SAR during plant–pathogen interactions [9,10,11]. In addition, an aromatic amino acid, serves as a precursor for various stress-related secondary metabolites, although its direct contribution to immune regulation remains to be fully elucidated [12].

SAR is characterized by the accumulation of salicylic acid (SA) and the subsequent activation of genes encoding pathogenesis-related proteins (PRs), such as *PR1* and *PR4*, resulting in enhanced resistance against biotrophic pathogens [13,14]. In contrast, ISR is predominantly regulated by jasmonic acid and ethylene signaling pathways [15], leading to the activation of defense-related genes, such as plant defensin 1.2 [16,17]. While microbial elicitors, including various *Bacillus* spp., have been extensively studied for their ability to trigger SAR or ISR, chemical elicitors such as *β*-aminobutyric acid have also been demonstrated to prime plant immune responses, particularly through the activation of SA-dependent signaling pathways [12,18]. These findings suggest that both microbial and chemical elicitors can effectively enhance plant immunity by modulating distinct hormonal pathways. Despite the increasing recognition of amino acids in plant defense responses, most previous studies have primarily focused on antimicrobial secondary metabolites produced by bacteria, with limited attention to the direct contributions of amino acids such as lysine.

*B. subtilis* M320 (BSM320), previously reported to suppress fungal diseases in *Lentinula edodes* (shiitake mushroom) [19], was observed to exert protective effects against cucumber powdery mildew. Comprehensive chemical analysis of BSM320 culture filtrates identified lysine as a major constituent. Based on these findings, we hypothesized that lysine may not act as a direct antimicrobial agent but rather as a priming molecule capable of activating plant defense responses. Therefore, the present study aimed to elucidate the defense-inducing mechanism of lysine, which is abundantly produced by BSM320. The outcomes of this study provide new insights into amino acid-mediated plant immunity and highlight lysine as a potential agent for the development of sustainable and environmentally friendly crop protection strategies.

## 2. Results

### 2.1. Identification of Antifungal Compounds from Culture Filtrate of BSM320

To identify the bioactive components responsible for suppressing cucumber powdery mildew (CPM), the culture filtrate of BSM320 was subjected to Diaion HP-20 column chromatography, and the pass fraction and acetone eluted fraction were separated for purification. CPM was suppressed on the acetone eluted fraction, but not on the pass fraction. Furthermore, The eluted fraction was extracted with ethyl acetate and then fractionated into an ethyl acetate. The aqueous phasethat had a protective effect on CPM was further analyzed by the following stepLiquid chromatography–mass spectrometry (LC–MS) analysis of the aqueous phase revealed the presence of lysine, phenylalanine, and tryptophan (Figure 1). Lysine was isolated as the major bioactive compound (fraction 23-H320 DW E3-3) through sequential purification, and its identity was confirmed by specific optical rotation measurement (+9.9°, c = 0.1, H_2_O). Previous studies have reported a key role for lysine and its catabolic derivatives, such as pipecolic acid, in enhancing plant defense responses [9,10,20]. High-performance liquid chromatography analysis further demonstrated that lysine was highly accumulated (8.4 mg/g) in BSM320-inoculated Luria–Bertani (LB) broth compared to the uninoculated control (Figure 2). In summary, lysine was identified as the principal compound produced by BSM320, suggesting its potential role in suppressing CPM.

### 2.2. Effect of Lysine Against Cucumber Powdery Mildew

To determine whether disease suppression was due to direct antifungal activity or the induction of host defense responses against cucumber powdery mildew, we evaluated the disease control efficacy of lysine at different concentrations [21]. Cucumber plants infected with powdery mildew were treated with different concentrations of lysine (500, 2500, and 5000 mg/L), and the suppression of disease symptoms was evaluated (Figure 3A). The disease control efficacies were 22% for distilled water (DW, negative control) and 44%, 92%, 99%, and 100% for lysine at 500, 2500, and 5000 mg/L and for Allstop (positive control), respectively (Figure 3B). These results indicate that higher concentrations of lysine (≥2500 mg/L) effectively suppressed the development of powdery mildew symptoms in cucumber plants. To determine whether lysine exerts a direct antifungal effect, spore germination assays were conducted. The DW control exhibited a spore germination rate of 32%, whereas treatment with 500 mg/L lysine resulted in a slightly higher germination rate of 47% (Figure 4). In contrast, treatments with 2500 and 5000 mg/L lysine yielded germination rates of 31% and 28%, respectively, with no significant differences compared to the DW control. However, treatment with Allstop (positive control, commercial biopesticide) substantially reduced conidial germination. These results suggest that lysine plays a role in plant defense, rather they imply that it does not act through germination inhibition.

### 2.3. Lysine-Induced Activation of Plant Defense Responses

Since lysine treatment did not directly inhibit spore germination, it was hypothesized to activate plant defense mechanisms (Figure 3 and Figure 4). To evaluate this, quantitative reverse transcription PCR (qRT-PCR) analysis was performed to assess the expression of defense-related genes, including *PR1*, *PR3*, lipoxygenase 1 (*LOX1*), lipoxygenase 23 (*LOX23*), WRKY transcription factor 20 (*WRKY20*), and L-type lectin receptor kinase 6.1 (*LecRK6.1*) [18,22,23]. Expression levels of these genes were significantly upregulated at 24 and 48 h following treatment with 2500 mg/L L-lysine compared to that with the control (Figure 5). These findings suggest that lysine treatment enhances the expression of defense-related genes, thereby contributing to improved resistance to cucumber powdery mildew.

The observed gene upregulation suggests that lysine may activate SAR pathways, ultimately enhancing plant immunity against fungal infections [9]. *PR1* is a well-established molecular marker of SAR and is strongly associated with antifungal defense responses. To further investigate whether lysine activates an SA-dependent SAR pathway, SA levels were quantified in cucumber leaves treated with varying concentrations of L-lysine (500, 2500, and 5000 mg/L) using LC–MS. SA levels increased proportionally with the lysine concentration, suggesting that lysine treatment promotes SA biosynthesis, a key step in the activation of SAR pathways (Figure 6). SA concentrations in lysine-treated leaves were 0.8, 1.0, and 1.2 mg/mL at 500, 2500, and 5000 mg/L lysine treatments, respectively (Figure 6). These results further support the role of lysine in inducing SA biosynthesis, leading to the activation of SAR. Taken together, these findings indicate that lysine functions as a priming agent by enhancing SA-dependent SAR and promoting resistance to fungal pathogens.

## 3. Discussion

The increasing demand for environmentally sustainable agriculture has accelerated the search for biologically derived crop protection strategies as alternatives to conventional chemical pesticides. Among various biologically derived agents, microbial culture filtrates have garnered significant attention due to their natural bioactive compounds, which enhance plant immunity with minimal environmental impact. Specifically, *B*. *subtilis* strains are known to produce diverse secondary metabolites with antifungal properties and plant growth-promoting effects. Considering these aspects, the culture filtrate from BSM320, previously identified as a potent antifungal agent, was investigated for its potential to control cucumber powdery mildew.

Our analysis of the BSM320 culture filtrate identified several amino acids, including lysine, phenylalanine, and tryptophan (Figure 1). Among these, lysine was isolated as the predominant bioactive compound responsible for antifungal activity (Figure 2). Consistent with prior studies, lysine has been reported to enhance disease resistance across multiple plant–pathogen interactions. For instance, lysine treatment effectively suppressed apple scab caused by *Venturia inaequalis*, apple spot disease caused by *Alternaria alternata*, and tomato bacterial wilt caused by *Ralstonia solanacearum* [9,24,25]. Moreover, lysine provided protection against rice blast disease caused by pathogens such as *Burkholderia glumae* and *Burkholderia plantarii* [26,27]. Similarly, cysteine has been shown to inhibit the growth and spore germination of pathogens like *Phaeomoniella* [28], further underscoring the significance of amino acids in plant defense. Our findings align with these reports, demonstrating a significant reduction in cucumber powdery mildew severity in cucumber plants treated with lysine, exhibiting a clear dose-dependent response (Figure 3). Notably, spore germination assays revealed no direct inhibitory effect of lysine on fungal conidial germination (Figure 4), suggesting that its protective mechanism primarily involves the induction of host plant defense responses rather than direct antifungal activity.

Recent studies have shown the importance of amino acid metabolism in regulating plant immune responses. Amino acids influence hormonal homeostasis, the C/N balance, and the biosynthesis of secondary metabolites, collectively contributing to the activation of plant defense pathways. Lysine, beyond its traditional role as a proteinogenic amino acid, also serves as a precursor to key signaling molecules such as pipecolic acid, a crucial regulator of SAR [11,27,29]. Although pipecolic acid levels were not directly quantified in this study, the significant upregulation of key defense-related genes (*PR1*, *PR3*, *LOX1*, *LOX23*, *WRKY20*, and *LecRK6*.1) coupled with elevated SA accumulation in lysine-treated cucumber plants strongly suggests lysine-mediated activation of SA-dependent SAR (Figure 5 and Figure 6).

Specifically, the increased expression of *PR1* and elevated SA concentrations collectively validate the hypothesis that lysine primes host immune responses to pathogen attack. Amino acids have been increasingly recognized for their roles in modulating defense signaling pathways, hormonal interactions, and plant–microbe interactions mediated by auxin-responsive *GH3* genes [30]. Additionally, lysine metabolism contributes to multiple plant stress responses and development [31]. Moreover, other amino acids, such as methionine, have been shown to upregulate genes like *9-LOX*, thus enhancing the biosynthesis of secondary metabolites crucial for plant defense [32].

Previous research also indicates that lysine can enhance the activities of defense-related enzymes such as β-1,3-glucanase and chitinase and stimulate *PR1* gene expression [4]. Lysine treatment has demonstrated broad-spectrum resistance in diverse plant systems, supporting our findings and highlighting its potential as an effective plant immunity elicitor [9,24,27]. Critically, although certain amino acids at high concentrations have direct antimicrobial effects, our results show that lysine primarily functions through host immune modulation rather than direct pathogen antagonism. This mechanism presents a sustainable approach for disease management, as it reduces selection pressure for pathogen resistance.

Collectively, our study provides compelling evidence for the role of lysine as an environmentally friendly priming agent capable of inducing SA-dependent SAR in cucumber plants. Amino acid-based priming thus represents a viable complement or alternative to chemical pesticides, aligning with sustainable agricultural practices [12,21]. Nevertheless, future studies should investigate the detailed molecular signaling cascades downstream of lysine perception, specifically clarifying whether lysine directly influences SA biosynthesis or functions through intermediates such as pipecolic acid or N-hydroxypipecolic acid [33]. Additionally, evaluating the efficacy of lysine-induced resistance against diverse pathogens and optimizing application protocols under field conditions remain critical steps toward practical agricultural implementation.

In conclusion, lysine produced by BSM320 emerges as a promising natural agent capable of enhancing cucumber resistance to powdery mildew through SA-mediated systemic resistance pathways. These findings enrich our understanding of amino acid-mediated plant immunity and offer a foundation for developing environmentally sustainable crop protection strategies.

## 4. Materials and Methods

### 4.1. Purification and Identification of Antimicrobial Compounds from BSM320 Culture Filtrate

BSM320 was inoculated in 10 L of LB broth and cultured at 28 °C with shaking at 120 rpm until the bacterial cell concentration reached 2 × 10^9^ CFU/mL. The culture broth was centrifuged at 6000 rpm for 15 min to remove bacterial cells, and the supernatant was filtered through a 0.45 μm membrane filter (Millipore, Billerica, MA, USA) to obtain the culture filtrate. The resulting filtrate was subjected to Diaion HP 20 column chromatography, and the pass fraction and acetone eluted fraction were separated for purification. The acetone fraction was extracted twice with an equal volume of ethyl acetate. The ethyl acetate and aqueous phases were separated, and the aqueous phase was subjected to further purification. For fractionation, the aqueous phase was adsorbed onto Diaion HP-20 (Mitsubishi Chemical, Tokyo, Japan) and Dowex 50WX8 (Sigma-Aldrich, St. Louis, MO, USA) resins. Sequential elution with DW and 1N ammonia yielded four fractions. Further purification of the active aqueous fraction was performed using medium-pressure liquid chromatography with cellulose resin, applying a gradient of isopropanol and 1N ammonia (ratios ranging from 0:10 to 3:7). This process resulted in the isolation of two subfractions, among which the major active component, designated as fraction 23-H320 DW E3-3, was identified. Structural characterization of the purified compound was performed by nuclear magnetic resonance (NMR) spectroscopy and electrospray ionization mass spectrometry (ESI–MS). The ^1^H-NMR spectrum exhibited characteristic proton signals at δH 3.33 (1H, triplet, J = 6.5 Hz) and δH 2.91 (1H, triplet, J = 7.5 Hz), while ESI-MS analysis revealed molecular ions at m/z 147 [M + H]^+^ and 169 [M + Na]^+^.

### 4.2. Plant Materials and Powdery Mildew Inoculation

Cucumber seedlings (*C*. *sativus* L. cv. Chunsim Baekdadagi; Dongwon Nongsan Seed Company, Yongin, Korea) were used in this study. Plants were grown in a controlled-environment chamber at 25 ± 2 °C under long-day conditions (16 h light/8 h dark) with 60% relative humidity until the development of 2–3 true leaves.

Powdery mildew-infected cucumber samples were collected from experimental fields at Hankyong National University (Anseong, Republic of Korea) and used as the source of natural inoculum. To promote infection, seedlings were exposed to field-derived powdery mildew spores, and successful infection was confirmed by visual observation of characteristic powdery lesions. For the foliar spray assay, cucumber seedlings were treated with different concentrations of L-lysine (500, 2500, and 5000 mg/L) by spraying the entire foliage with 3 d intervals until runoff. DW was used as the negative control, and a commercial microbial fungicide, Allstop (KoreaBio, Hwaseong, Republic of Korea), served as the positive control. The cucumber plants were maintained in the plastic box including the cucumber plants infected by powdery mildew at 25 ± 2 °C under long-day conditions (16 h light/8 h dark) with 60% relative humidity. Ten plants were used for each treatment with three replicates. Disease severity was evaluated 12 d after the third treatment. The disease index was scored on a scale from 0 to 5 based on the percentage of infected leaf area: 0 = no visible symptoms; 1 = 0.1–5% infection; 2 = 5.1–20%; 3 = 20.1–40%; 4 = 40.1–60%; and 5 = 60.1–100% infection. Disease severity was calculated to reliably quantify phenotypic data using the following equation:$$Disease\;severity\;(\%)=\frac{\Sigma (Disease\;index\times Number\;of\;infected\;leaves\;at\;each\;rating)}{4N}\times 100$$
where N = number of leaves assessed.

### 4.3. Spore Germination Rate of Cucumber Powdery Mildew Fungus

To assess the direct antifungal activity of lysine, spore germination assays were conducted using powdery mildew spores collected from naturally infected cucumber leaves. The collected spores were suspended in sterile DW and adjusted to a concentration of 2 × 10^4^ spores/mL. Agar blocks were prepared by mixing 2% agar with each treatment solution and pouring the mixture onto microscope slides (Paul Marienfeld GmbH & Co. KG, Lauda-Königshofen, Germany). The treatment groups included DW (negative control), Allstop (positive control), and L-lysine at concentrations of 500, 2500, and 5000 mg/L. A 20 µL aliquot of the prepared spore suspension was placed onto the surface of each agar block, and a coverslip was gently placed on top. The slides were incubated in a moist chamber at 25 °C under saturated humidity and continuous fluorescent illumination for 48 h to promote spore germination. After incubation, spore germination was assessed under a light microscope by randomly selecting and observing at least 100 spores per treatment. A spore was considered germinated if the germ tube length was equal to or greater than the diameter of the spore.

### 4.4. Quantitative Reverse Transcription PCR (qRT-PCR)

qRT-PCR analysis was performed to evaluate the expression of defense-related genes in cucumber plants treated with L-lysine. Four-week-old cucumber plants were foliar spray-treated with L-lysine at a concentration of 2500 mg/L. Leaf samples were collected at 0, 24, 48, and 72 h after treatment. Total RNA was extracted from the leaves using the Favorgen RNA Extraction Kit (Favorgen Biotech Corp., Vienna, Austria) according to the manufacturer’s instructions. To eliminate genomic DNA contamination, RNA samples were treated with RNase-free DNase I (Thermo Fisher Scientific, Waltham, MA, USA). First-strand cDNA was synthesized from 1 μg of DNA-free RNA using HelixCript™ Easy Reverse Transcriptase (NanoHelix, Daejeon, Republic of Korea). qRT-PCR reactions were carried out in a 20 μL volume containing 10 μL of Dyne qPCR 2X PreMix (DyneBio, Seongnam, Republic of Korea), 1 μL of cDNA template, and 10 pmol of each specific primer. The amplification protocol consisted of an initial denaturation at 95 °C for 15 s, followed by 40 cycles of denaturation at 95 °C for 15 s, annealing at 58 °C for 30 s, and extension at 72 °C for 30 s. Primer sequences for the defense-related genes are listed in Appendix A. Gene expression levels were analyzed using the LightCycler^®^ 96 System (Roche, Basel, Switzerland) and normalized to the expression of the internal reference gene (actin).

### 4.5. Extraction and Measurement of SA from Cucumber Leaves

Cucumber leaves were treated with L-lysine at concentrations of 500, 2500, and 5000 mg/L. After 72 h, leaf tissues were harvested and immediately frozen in liquid nitrogen. The frozen samples (10 g) were ground into a fine powder and mixed with 90 g of silicon dioxide. SA was extracted by adding 30 mL of 90% methanol to the mixture. The extracts were concentrated under reduced pressure using a rotary evaporator. The resulting residues were resuspended in 1 mL of 5% (*w*/*v*) trichloroacetic acid (Sigma-Aldrich, St. Louis, MO, USA) and 10 mL of 99.8% methanol. The final volume of each sample was adjusted to 50 mL with DW, followed by centrifugation at 8000 × g for 10 min. The supernatants were collected and subjected to LC–MS analysis for SA quantification. LC–MS analysis was performed using a Kinetex C18 column (2.6 μm, 100 mm × 2.1 mm; Phenomenex, Torrance, CA, USA). The calibration curve for SA quantification ranged from 10 to 500 ng/mL, with a coefficient of determination (R^2^) greater than 0.99, ensuring a high accuracy and precision.

### 4.6. Statistical Analysis

All experiments were performed with at least three independent biological replicates. Data are presented as the mean ± standard deviation. Statistical significance among treatment groups was evaluated by one-way analysis of variance, followed by Tukey’s multiple comparison test. Differences were considered statistically significant at *p* < 0.05. All statistical analyses were performed using SAS software version 7.1 (SAS Institute Inc., Cary, NC, USA).

## 5. Conclusions

This study has demonstrated that L-lysine treatment enhances cucumber resistance against powdery mildew not through direct inhibition of fungal spore germination but by activating plant immune responses. The upregulation of key defense-related genes, including *PR1*, *PR3*, and *LOX1*, along with the increased accumulation of SA strongly suggests that L-lysine triggers SAR pathways. These findings highlight the potential of L-lysine as a natural priming agent for enhancing plant immunity, offering a sustainable and environmentally friendly strategy for disease management. Further research is required to elucidate the detailed molecular mechanisms underlying lysine-mediated defense activation and to assess its practical applicability across different crops and under field conditions.

## Figures and Tables

**Figure 1 ijms-26-06882-f001:**
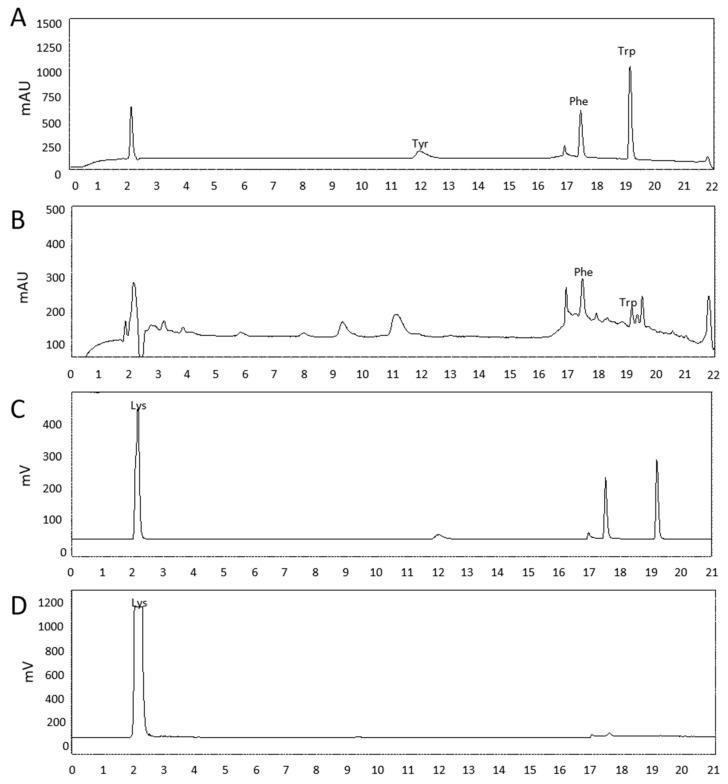
Identification of amino acids in the fraction of *Bacillus subtilis* M320 (BSM320) culture filterate by HPLC analysis (**A**) HPLC-UV, amino acid standards solution (lysine, tryptophan, tyrosine, phenylalanine), (**B**) HPLC-UV, fraction of BSM320 culture filterate, (**C**) HPLC-ELSD, amino acid standard solution, (**D**) HPLC-ELSD, fraction of BSM320 culture filterate. lysine (Lys), phenylalanine (Phe), tyrosine (Tyr), and tryptophan (Trp).

**Figure 2 ijms-26-06882-f002:**
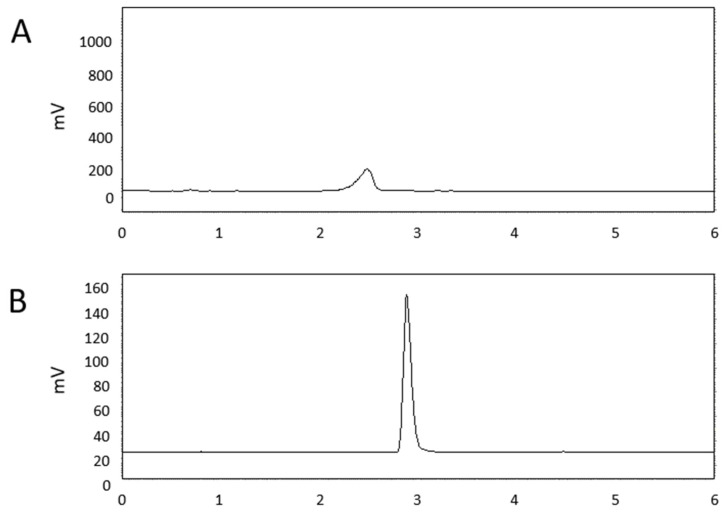
Quantification of lysine production by BSM320 using high-performance liquid chromatography (HPLC) analysis. (**A**) HPLC chromatogram of Luria–Bertani (LB) medium without BSM320 inoculation. (**B**) HPLC chromatogram of LB medium after cultivation with BSM320. The y-axis represents the detector response (mAU), and the x-axis indicates the retention time (min).

**Figure 3 ijms-26-06882-f003:**
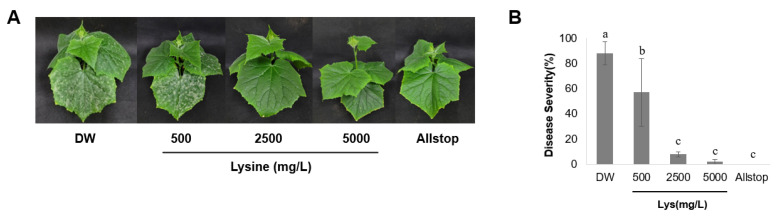
Control effect of L-lysine on cucumber powdery mildew. (**A**) Representative images of cucumber leaves treated with different concentrations of L-lysine (500, 2500, and 5000 mg/L) or a commercial product (Allstop, a biopesticide). Distilled water (DW) was used as the untreated control. (**B**) Disease severity was evaluated at 7 d post-inoculation, expressed as the percentage of leaf area covered with powdery mildew lesions. Bars represent the mean ± standard deviation (SD) of three independent replicates. Different lowercase letters above the bars (a, b, c) indicate statistically significant differences (*p* < 0.05), as determined by one-way analysis of variance (ANOVA) followed by Tukey’s multiple comparison test.

**Figure 4 ijms-26-06882-f004:**
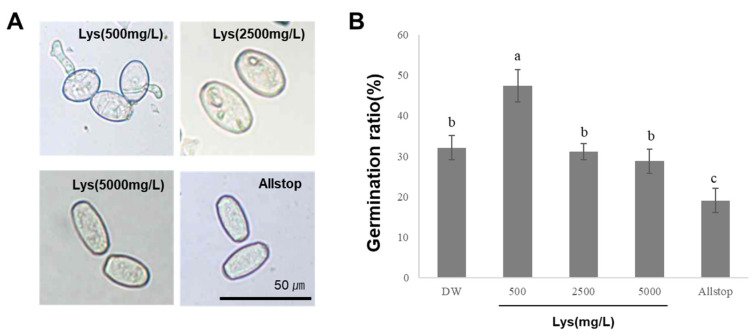
Effects of L-lysine treatment on conidial germination of *Sphaerotheca fusca*. (**A**) Light microscopy images of *S. fusca conidia* treated with L-lysine at 500, 2500, and 5000 mg/L or a commercial positive control (Allstop, a biopesticide). DW was used as the untreated control. (**B**) Germination rates of *S. fusca* conidia were assessed after 48 h. A conidium was considered germinated if the germ tube was equal to or longer than the diameter. Bars represent the mean ± SD from three independent replicates. Different lowercase letters above the bars indicate statistically significant differences (*p* < 0.05), as determined by one-way ANOVA followed by Tukey’s post hoc multiple comparison test.

**Figure 5 ijms-26-06882-f005:**
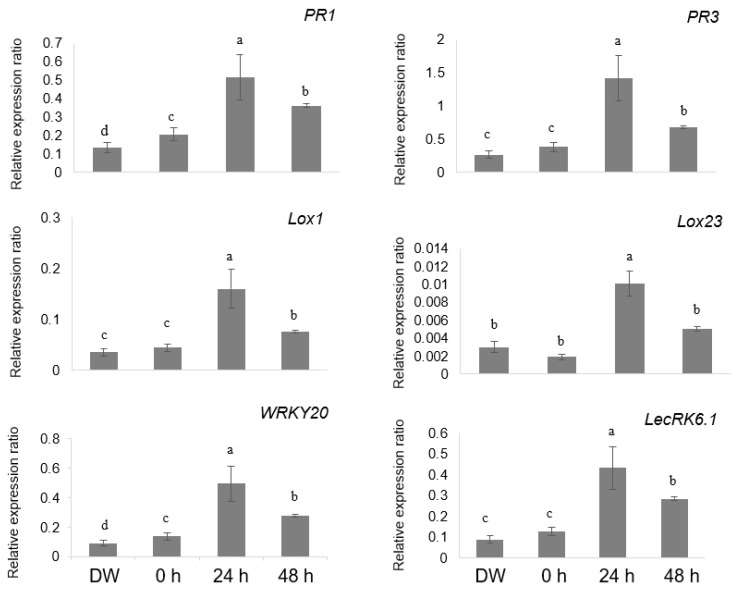
Quantitative reverse transcription PCR analysis of defense-related gene expression in cucumber plants treated with lysine. Relative expression levels of defense-related genes—pathogenesis-related proteins 1 and 3 (*PR1* and *PR3*), lipoxygenase 1 and 23 (*LOX1* and *LOX23*), WRKY transcription factor 20 (*WRKY20*), and L-type lectin receptor kinase 6.1 (*LecRK6*.1)—in cucumber leaves treated with lysine 2500 mg/L over a 48 h period. DW was used as the negative control, and “0 h” indicates the sampling time immediately prior to lysine application. The y-axis represents relative fold change in gene expression. Different lowercase letters above each bar indicate statistically significant differences as determined by one-way ANOVA followed by Tukey’s post hoc multiple comparison test (*p* < 0.05). Bars represent the mean ± SD from three independent biological replicates.

**Figure 6 ijms-26-06882-f006:**
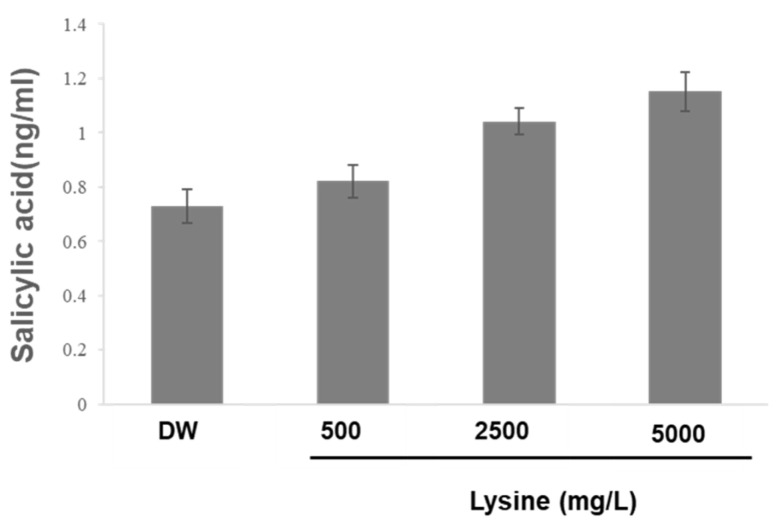
Salicylic acid (SA) accumulates in cucumber leaves following lysine treatments. SA levels were quantified in cucumber leaves treated with 500, 2500, and 5000 mg/L of L-lysine via foliar spray. Leaves were harvested 72 h post-treatment, and SA concentrations were measured using LC–MS. Bars represent the mean ± SD of three independent biological replicates.

## Data Availability

The data presented in this study are available on request from the corresponding author.

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
