# Peer review of "L-Lysine from Bacillus subtilis M320 Induces Salicylic-Acid–Dependent Systemic Resistance and Controls Cucumber Powdery Mildew"

_ijms, 2025, doi:10.3390/ijms26146882_

Round 1
Reviewer 1 Report
Comments and Suggestions for Authors
The manuscript of Ja-Yoon Kim and Hee-Wan Kang, submitted to International Journal of Molecular Sciences (ijms-3692843) investigates the use of lysine against the cucumber powdery mildew fungus Podosphaera xanthii.
In my opinion, the results are important, and the work was conducted decently.
The presentation of findings is mostly clear; however the manuscript is less well-written. In general, the text is somewhat repetitive, with several redundancies, thus, a streamlining and shortening the text would be welcome. For example, in the results section, several details from the materials and methods section are unnecessarily repeated.
Altogether, I suggest minor revisions to be made in the manuscript before publication.
There are several minor concerns that the authors have to consider and correct in order to improve the manuscript. These I have listed below.
Line 33. As far as I know, Podosphaera fusca does not infect cucumbers, as it is confined to plant hosts of the genus Doronicum (Braun and Cook 2011, Taxonomic Manual of the Erysiphales). Other authors treat the two names as synonyms. Please use cautious wording here. I would mention P. xanthii only in the introduction, as an example.
In addition:
Most probably, judged by the morphology of conidia, the authors are dealing with Podosphaera xanthii during the work. However, the sequencing of the internal transcribed spacer (and then submission of the sequence to the GenBank) would be needed to avoid potential confusion and taxonomic constraints associated with the cucumber powdery mildew fungus.
Line34: delete “(-relate”, I suppose.
Line 42: ‘have’ instead of ‘has’
l78-90: In my view, a summary of the text at the end of the introduction is not needed. Instead, a clear and short, streamlined description of aims would be sufficient.
Line132: I do not think that these results would indicate that lysine plays a role in plant defense, rather they imply that it does not act through germination inhibition. Please revise the wording.
Line 188 - Figure 6: Salicylic acid content of control leaves (untreated or dw treated) are not given, thus the measured amounts for lysine treatments cannot be properly evaluated without a reference amount. Please address this situation.
Line 283: How many plants were used in the tests? Sample numbers need to be added to the text.
Line 274 says “active aqueous fraction”. Does this mean that all the fractions were tested? The text does not reveal how the authors found out which fraction is the one with activity. The process of investigation of different fractions and biological activity needs to be clearly described.
L297-298 the text says “Disease severity was evaluated 12 d after the third treatment. Disease severity was assessed 4 d after treatment” This seems to be a contradiction. Please clarify how many treatments were done and when the scoring was conducted.
L303: Why did you divide by 4 in the equation (although there were 5 symptom categories)?
L337 – Table 1. You have to add references and refer to papers from which the primers were taken.
Results using CsPAL and CsCupi4 are not presented. Were these primers used? If not, please remove them from the list and make sure there are no unused primers listed.
Author Response
Comments 1: Line 33. As far as I know, Podosphaera fusca does not infect cucumbers, as it is confined to plant hosts of the genus Doronicum (Braun and Cook 2011, Taxonomic Manual of the Erysiphales). Other authors treat the two names as synonyms. Please use cautious wording here. I would mention P. xanthii only in the introduction, as an example.
Response 1: Thank you for pointing this out. We agree with this comment. Therefore, Mention exactly where in the revised manuscript this change can be found page number 1 and line 11
Comments 2: In addition:
Most probably, judged by the morphology of conidia, the authors are dealing with Podosphaera xanthii during the work. However, the sequencing of the internal transcribed spacer (and then submission of the sequence to the GenBank) would be needed to avoid potential confusion and taxonomic constraints associated with the cucumber powdery mildew fungus.
Response 2: Thank you for your additional comments. I totally agree with you.
Comments 3: Line34: delete “(-relate”, I suppose.
Response 3: This change can be found page number 1 and lines 33.
Comments 4: Line 42: ‘have’ instead of ‘has’
Response 4: This change can be found in line 41 of revised manuscript
Comments 5: l78-90: In my view, a summary of the text at the end of the introduction is not needed. Instead, a clear and short, streamlined description of aims would be sufficient.
Response 5: We have, accordingly, revised to emphasize this point. This change can be found – page number2, paragraph 5, and lines 75-84.
Comments 6: Line132: I do not think that these results would indicate that lysine plays a role in plant defense, rather they imply that it does not act through germination inhibition. Please revise the wording.
Response 6: Thank you for kind your comments. This change can be found – page number 4, paragraph 1, and lines 129-131 of the revised manuscript.
Comments 7: Line 188 - Figure 6: Salicylic acid content of control leaves (untreated or dw treated) are not given, thus the measured amounts for lysine treatments cannot be properly evaluated without a reference amount. Please address this situation.???
Response 7: Thank you for the comment. The SA content of DW-treated control leaves has been included in the revised Figure 6. This change can be found Figure 6 between lines 184 and 185 of page number 6.
Comments 8: Line 283: How many plants were used in the tests? Sample numbers need to be added to the text.
Response 8: Thank you for your comments. Ten cucumber seedlings were used for each treatment with three replicates. The revised content can be found in page number 9 and lines 296-2927of 4-2 Materials and Methods.
Comments 9: Line 274 says “active aqueous fraction”. Does this mean that all the fractions were tested? The text does not reveal how the authors found out which fraction is the one with activity. The process of investigation of different fractions and biological activity needs to be clearly described.
Response 9: Thank you for your comment. "
In Materials and Methods, This change can be found in page number 8, and lines 265-267. Additional change can be found In Results, page numbers 2 and 3, and lines 87-96. We purified the active ingredients after checking whether the fractions suppress CPM..
Comments 10: L297-298 the text says “Disease severity was evaluated 12 d after the third treatment. Disease severity was assessed 4 d after treatment” This seems to be a contradiction. Please clarify how many treatments were done and when the scoring was conducted.
Response 10: Thank you for your comment. Disease severity is 12 days after the third treatment, The change can be found in page number 9 and line 290-297.
Comments 11: L303: Why did you divide by 4 in the equation (although there were 5 symptom categories)?
Response 11: Thank you for your comment. You are correct there were 5 symptom categories, and the equation has been revised to divide by 5 accordingly in the main text.
Comments 12: L337 – Table 1. You have to add references and refer to papers from which the primers were taken. Results using CsPAL and CsCupi4 are not presented. Were these primers used? If not, please remove them from the list and make sure there are no unused primers listed.
Response 12: Thank you for your comment. CsPAL and CsCupi4 primers were not used in this study and have been removed from Table 1 accordingly. We have ensured that all primers listed are those actually used in the experiments. References for each primer have been added to the table, and all cited primer sequences are documented in reference 18 (Kim and Kang, Genes, 2023). This reference has also been added to the main text.

Reviewer 2 Report
Comments and Suggestions for Authors
Dear Author,
This study investigated the mechanism by which L-lysine produced by Bacillus subtilis M320 (BSM320) enhances the resistance of cucumber to powdery mildew through activating the SA-dependent systemic acquired resistance (SAR) pathway. The design of the study was reasonable, and the data supported the conclusion. It provided new insights into the amino acid-mediated plant immunity mechanism and has potential for agricultural applications. The experimental methods were rigorous, and the results were presented clearly, but some details need to be improved. Final decision: Accept after Minor Revisions.
The main revision suggestions are as follows:
1. Chart annotations: The Y-axis label of Figure 3B, "Disease Severity (%)", should be clearly stated as "Disease severity (%)"; the horizontal axis of Figure 5 should be reconsidered and either be supplemented with explanations for each table or integrated into one figure.
2. It is suggested to upload the content of Table 1 as an attachment rather than including it in the main text.
3. In the discussion section, it is recommended to add subheadings to avoid excessive repetition of the descriptions in the results section.
Date: June 18th, 2025
Author Response
The main revision suggestions are as follows:
Comments 1: Chart annotations: The Y-axis label of Figure 3B, "Disease Severity (%)", should be clearly stated as "Disease severity (%)"; the horizontal axis of Figure 5 should be reconsidered and either be supplemented with explanations for each table or integrated into one figure.
Response 1:
Thank you for your suggestion. The Y-axis label of Figure 3B has been revised to "Disease severity (%)" as recommended. For Figure 5, the horizontal axis has been adjusted for consistency. These changes have been applied in the revised manuscript.
Comments 2: It is suggested to upload the content of Table 1 as an attachment rather than including it in the main text.
Response 2: Thank you for your suggestion. Table 1 has been removed from the main text and provided instead as a supplementary file (Supplementary Table 1). The reference to Table 1 in the main text has been updated accordingly.
Comments 3: In the discussion section, it is recommended to add subheadings to avoid excessive repetition of the descriptions in the results section.
Response 3: Thank you for your suggestion.
Date: June 18th, 2025
